# Multi-value Rule Sets for Interpretable Classification with Feature-Efficient Representations

**Tong Wang**
Tippie School of Business
University of Iowa
Iowa City, IA 52242
`tong-wang@uiowa.edu`

## Abstract

We present the Multi-value Rule Set (MRS) for interpretable classification with feature efficient presentations. Compared to rule sets built from single-value rules, MRS adopts a more generalized form of association rules that allows multiple values in a condition. Rules of this form are more concise than classical single-value rules in capturing and describing patterns in data. Our formulation also pursues a higher efficiency of feature utilization, which reduces possible cost in data collection and storage. We propose a Bayesian framework for formulating an MRS model and develop an efficient inference method for learning a maximum *a posteriori*, incorporating theoretically grounded bounds to iteratively reduce the search space and improve the search efficiency. Experiments on synthetic and real-world data demonstrate that MRS models have significantly smaller complexity and fewer features than baseline models while being competitive in predictive accuracy. Human evaluations show that MRS is easier to understand and use compared to other rule-based models.

## 1 Introduction

In many real-world applications of machine learning, human experts desire the interpretability of a model as much as the predictive accuracy. As opposed to "black box" models, interpretable models are easy for humans to understand and extract insights, which is imperative in domains such as healthcare, law enforcement, etc. In some occasions, the need for interpretability even outweighs that for accuracy due to legal or ethnic concerns. Among different forms of interpretable models, we are particularly interested in rule-based models in this paper. This type of models produce decisions based on a set of rules following simple "if-else" logic: if a rule (or a set of rules) is satisfied, the model outputs the corresponding decision. The set of rules can be either ordered [17, 35, 5] or unordered [15, 30, 19, 24], depending on the specific model structure.

Prior rule-based models in the literature are from built single-value rules [15, 30, 19]. For example, [State = California] *AND* [Marital status = married], where a *condition* (e.g., [state=California]) is a pair of a feature (e.g., state) and a single value (e.g., California). However, while single-value rules can express primitive concepts, they are inadequate in capturing more general trends in the underlying data, especially when working with features with a medium to high cardinality. Rules built from these features tend to have too small support. They are either less likely to be selected in the final output, introducing selection bias in the model [7], or induce a large model if selected, hurting the model interpretability. For example, to capture a set of married or divorced people who live in California, Texas, Arizona, and Oregon, a model needs to include eight rules, each rule being a combination of a state and a marital status, yielding an overly complicated model. As modern machine learning has in part moved on to pursue a better and more concise way of model

presentation as well as predictive accuracy, single-value rules do not suffice for some applications, neither do models built from them.

To mitigate this problem, rules in a more generalized form have been proposed in the literature that allow multiple values [26, 22], also called internal disjunctions of values in a condition [6]. For example: [state = California or Texas or Arizona or Oregon] *AND* [marital status = married or divorced]. In this case, we only need one rule instead of eight single-value rules, yielding a more concise presentation while preserving the information. We refer to rules of this form multi-value rules, which will serve as the building blocks of our proposed model in this paper. The prior efforts on multi-value rules have mainly focused on finding individual rules and using heuristics such as interestingness, confidence, etc., instead of building a principled classification model with a global objective function that considers predictive accuracy and model complexity.

Another important aspect that has been overlooked by previous rule-based models is the need to control the total number of unique features. The number of different entities humans need to comprehend is directly associated with how easy it is to understand the model, as confirmed by the conclusions of Miller [21] relative to the magical number seven. With fewer features involved, it also becomes easier for domain experts to gain clear insights into the data. In practice, models using fewer features are easier to understand and bring down the overall cost in data collection.

To combine the factors considered above, we propose a novel rule-based classifier, *Multi-value Rule Set* (MRS), which is a set of multi-value rules. An instance is classified as positive if it satisfies at least one of the rules. A MRS has great advantages over models built from single-value rules in (i) a more concise presentation of information and (ii) using a smaller number of features in the model.

We develop a Bayesian framework for learning MRS which provides a unified framework to jointly optimize data fitting and model complexity without directly "hard" controlling either. We propose a principled objective combining the interpretability and the predictive accuracy where we devise a prior model that promotes a *small* set of *short* rules using a *few* features. We propose an efficient inference algorithm for learning a maximum *a posteriori* model. We show with experiments on standard data sets that MRS produces predictive accuracy comparable to or better than prior art with lower complexity and fewer overall features.

## 2   Related Work

There has been a series of research on developing rule-based models for classification [31, 15, 12, 3, 28, 19, 5, 24, 24]. Various structures and formats of models were proposed, from the earlier work on Classification based on Association Rules (CBA) [18] and Repeated Incremental Pruning to Produce Error Reduction (Ripper) [5] to more recent work on rule sets [31, 15, 19] and rule lists [16, 29]. A major development along this line of work is that interpretability has been recognized and emphasized. Therefore controlling model complexity for easier interpretation is becoming an important component in the modeling. However, previously mentioned models rely on single-value rules and are limited in the expressive power, leaving redundancy in the model. In addition, learning in previous methods is mostly a two-step procedure[31, 18, 16], that first uses off-the-shelf data mining algorithms to generate a set of rules and then chooses a set from them to form the final model. This in practice will encounter the bottleneck of mining rules of a large maximum length (millions of rules can be generated from a medium size data set if the maximum length is set to only 3 [31]). Furthermore, few of the previous works consider limiting the number of features. Our model aims to combines rule learning and feature assignment into the same process.

Our work is broadly related to generalized association rules that consider disjunctive relationships. Among various works along this line, some consider disjunction in the rule level, using the disjunction connector instead of a conjunction connector as used by classical rule form. For example, $a_1 \vee a_2 \vee \cdots \to Y$, where $a_1$ is a rule. Representative works include [10, 9, 8]. This primitive form of rules was extended to consider disjunctions in the condition / literal level [22], yielding multi-value rules of the form $(a_1 \vee a_2 \vee \cdots) \wedge (b_1 \vee b_2 \cdots) \to Y$. Prior efforts have mainly focused on mining individual multi-value rules [25, 10] using heuristics such as interestingness. Some works built classifiers comprised of multi-value rules [1, 2, 20]. However, they still rely on greedy methods such as greedy induction to build a model, and they do not consider model complexity or restrict the number of features. Here, we optimize a global objective that considers predictive accuracy, model size, and the total number of features. By tuning the parameters in the Bayesian framework, our model can

strike a nice balance between the different aspects of the model to suit the domain specific need of users.

# 3 Multi-value Rule Sets

We work with standard classification data set $S$ that consists of $N$ observations $\{\mathbf{x}_n, y_n\}_{n=1}^N$. Let $\mathbf{y}$ represent the set of labels. Each observation has $J$ features, and we denote the $j^{\text{th}}$ feature of the $n^{\text{th}}$ observation as $x_{nj}$. Let $\mathcal{V}_j$ represent a set of values the $j^{\text{th}}$ feature takes. This notation can be adapted to continuous attributes by discretizing the values.

## 3.1 Multi-value Rules

Now we introduce the basic components in Multi-value Rule Set model.

**Definition 1** *An item is a pair of a feature $j$ and a value $v$, where $j \in \{1, 2, \cdots, J\}$ and $v \in \mathcal{V}_j$.*

**Definition 2** *A condition is a collection of items with the same feature $j$, denoted as $c = (j, V)$, where $j \in \{1, 2, \cdots, J\}$ and $V \subset \mathcal{V}_j$. $V$ is a union of values in the items.*

**Definition 3** *A multi-vale rule is a conjunction of conditions, denoted as $r = \{c_k\}_k$.*

Interchangeable values are grouped into a value set in a condition, such as [state = California or Texas or Arizona or Oregon]. Following the definitions, an item is the atom in a multi-value rule. It is also a special case of a condition with a single value, for example, [state = California].

Now we define a classifier built from multi-value rules. By an abuse of notation, we use $r(\cdot)$ to represent a Boolean function that indicates if an observation satisfies rule $r$: $r(\cdot) : \mathcal{X} \to \{0, 1\}$. Let $R$ denote a Multi-value Rule Set. We define a classifier $R(\cdot)$:

$$R(\mathbf{x}) = \begin{cases} 1 & \exists r \in R, r(\mathbf{x}) = 1 \\ 0 & \text{otherwise.} \end{cases} \tag{1}$$

$\mathbf{x}$ is classified as positive if it satisfies *at least* one rule in $R$ and we say $\mathbf{x}$ is *covered* by $r$.

## 3.2 MRS Formulation

Our proposed framework considers two aspects of a model: 1) interpretability, characterized by a prior model for MRS, which considers the complexity (number of rules and lengths of rules) and feature assignment. 2) predictive accuracy, represented by the conditional likelihood of data given an MRS model. Both components have tunable parameters to trade off between interpretability and predictive accuracy. Now we formulate the model.

**Prior for MRS** The prior model for MRS jointly determines the number of rules $M$, lengths of rules $\{L_m\}_{m=1}^M$ and feature assignment $\{z_m\}_{m=1}^M$, where $m$ is the rule index. We propose a two-step process for constructing a rule set, where the first step determines the size and shape of an MRS model and the second step fills in the empty "boxes" with items.

*Creating empty "boxes" - complexity assignment:* First, we draw the number of rules $M$ from a Poisson distribution, where $\lambda_M \sim \text{Gamma}(\alpha_M, \beta_M)$. Second, we determine the number of items in each rule, denoted as $L_m$. $L_m \sim \text{Poisson}(\lambda_L)$, which is a Poisson distribution truncated to only allow positive outcomes. The arrival rate for this Poisson distribution, $\lambda_L$, is governed by a Gamma distribution with parameters $\alpha_L, \beta_L$. Since we favor simpler models for interpretability purposes, we set $\alpha_L < \beta_L$ and $\alpha_M < \beta_M$ to encourage a small set of short rules. These two steps together determine the size and shape of an MRS model. Therefore, we call parameters $\alpha_M, \beta_M, \alpha_L, \beta_L$ shape parameters. $H_s = \{\alpha_M, \beta_M, \alpha_L, \beta_L, \theta\}$. This step creates empty "boxes" to be filled with items in the following step and assigns overall complexity to the model.

*Filling "boxes" - feature assignment:* A $m$-th rule is a collection of $L_m$ "boxes", each containing an item. Let $z_{mk}$ represent the feature assigned to the $k^{\text{th}}$ box in the $m$-th rule, where $z_{mk} \in \{1, ..., J\}$ and $z_m$ represent the set of feature assignments in the $m$-th rule. We sample $z_m$ from a multinomial distribution with weights $p$ drawn from a Dirichlet distribution parameterized by

hyperparameter $\theta = \{\theta_j\}_{j=1}^J$. Let $l_{mj}$ denote the number of items with attribute $j$ in the $m$-th rule, i.e., $l_{mj} = \sum_k \mathbb{1}(z_{mk} = j)$ and $\sum_j l_{mj} = L_m$. It means $l_{mj}$ items share the same feature $j$ and therefore can be merged into a condition. We truncate the multinomial distribution to only allow $l_{mj} \leq |\mathcal{V}_j|$. Remarks: we use Multinomial-Dirichlet distribution for feature assignment for its clustering property of the outcomes. The prior model will tend to re-use features already in the rule. This is consistent with the interpretability goal of our model: we would like to form a MRS model with fewer features so that multiple items can be merged in to one condition. The prior does not consider values in each item since they do not affect the size and the shape of the model and therefore have no effect on the interpretability. In summary, the prior for MRS model follows a distribution below, where $C$ is a function of $H_s$ and $\Gamma(\cdot)$ is a gamma function.

$$p(R; H_s) \propto \frac{\Gamma(M^* + \alpha_M)C^M}{\Gamma(M^* + 1)} \prod_{m=1}^M \frac{\Gamma(L_m + \alpha_L)}{\Gamma(L_m + 1)} \frac{\prod_{j=1}^J \Gamma(l_{mj} + \theta_j)}{(\beta_L + 1)^{L_m}\Gamma(L_m + \sum_{j=1}^J \theta_j)}. \tag{2}$$

**Conditional Likelihood** Now we consider the predictive accuracy of a MRS by modeling the conditional likelihood of labels $\mathbf{y}$ given features $\mathbf{x}$ and a MRS model $R$. Our prediction on the outcomes are based on the coverage of MRS. According to formula (1), if an observation satisfies $R$ (covered by $R$), it is predicted to be positive, otherwise, it's negative. We assume label $y_n$ is drawn from a Bernoulli distribution with probabilities $\rho_+$ or $\rho_-$ to be consistent with the predicted outcome. Specifically, when $R(\mathbf{x}_n) = 1$, i.e., $\mathbf{x}_n$ satisfies the rule set, $y_n$ has probability $\rho_+$ to be positive, and when $R(\mathbf{x}_n) = 0$, $y_n$ has probability $\rho_-$ to be negative. $\rho_+, \rho_-$ govern the predictive accuracy on the training data. We assume that they are drawn from two Beta distributions with hyperparameters $(\alpha_+, \beta_+)$ and $(\alpha_-, \beta_-)$, respectively, which control the predictive power of the model. The conditional likelihood is shown below, given parameters $H_c = \{\alpha_+, \beta_+, \alpha_-, \beta_-\}$:

$$p(\mathbf{y}|\mathbf{x}, R; H_c) \propto B(\text{TP} + \alpha_+, \text{FP} + \beta_+)B(\text{TN} + \alpha_-, \text{FN} + \beta_-), \tag{3}$$

where TP, FP, TN and FN represent the number of true positives, false positives, true negatives and false negatives, respectively. $B(\cdot)$ is a Beta function which comes from integrating out $\rho_+, \rho_-$ in the conditional likelihood function.

We will write $p(R; H_s)$ as $p(R)$ and $p(\mathbf{y}|\mathbf{x}, R; H_c)$ as $p(\mathbf{y}|\mathbf{x})$, ignoring dependence on parameters when necessary. Regarding setting hyperparameters $H_s, H_c$, there are natural settings for $\theta$ (all entries being 1). This means there's no prior preference for features. For Gamma distributions, we set $\alpha_M$ and $\alpha_L$ to 1. Then the strength of the prior for constructing a simple MRS depends on $\beta_M$ and $\beta_L$. Increasing $\beta_M$ and $\beta_L$ decreases the expected number of rules and the expected length of rules, thus penalizing more on larger models. There are four real-valued parameters in the conditional likelihood to set, $\alpha_+, \beta_+, \alpha_-, \beta_-$. They jointly control the probability that a prediction of MRS model is correct. Therefore we should always set $\alpha_+ > \beta_+, \alpha_- > \beta_-$. The ratios of $\alpha_+, \beta_+$ and $\alpha_-, \beta_-$ are associated with the expected predictive accuracy. Setting values of the parameters can be done through cross-validation, another layer of hierarchy with more diffuse hyperparameters, or plain intuition.

### 3.3 Clustering of Features

We use Multinomial-Dirichlet in the prior model to take advantage of the "clustering" effect in feature assignment. Our goal is to formulate a model which favors rules with fewer features. Here we prove this effect. Let $R$ denote a MRS model and $l_{mj}$ represent the number of items in rule $m$ taking feature $j$. Now we do a small change in $R$: pick two features $j_1, j_2$ in rule $m$ where $l_{mj_1} \geq l_{mj_2}$ and replace an item taking feature $j_2$ with an item taking feature $j_1$, and we denote the new rule set as $R'$. Every rule in $R'$ remains the same as $R$ except in the $m$-th rule. Let $l'_{mj_1}, l'_{mj_2}$ denote the number of items taking feature $j_1$ and $j_2$ in the new model, and $l'_{mj_1} = l_{mj_1} + 1$ and $l_{mj_2} = l'_{mj_2} - 1$. We claim this flip of feature increases the prior probability of the model, i.e.,

**Theorem 1** *If $l_{mj_1} + \theta_{j_1} \geq l_{mj_2} + \theta_{j_2}$, then $p(R') \geq p(R)$.*

When we choose uniform prior where all $\theta_j$ are equal, the theorem will be reduced to a simpler form, that the model always tends to reuse the most prevalent features. For example, given two rules [state = California or Texas] *AND* [marital status = married] and [state = California] *AND* [marital status = married] *AND* [age $\geq$ 45], our model will favor rule sets containing the former, if everything else being equal. (All proofs are in the supplementary material.)

# 4   Inference Method

Inference for rule-based models is challenging because it involves a search over exponentially many possible sets of rules: since each rule is a conjunction of conditions, the number of rules increases exponentially with the number of features in a data set, and the solution space (all possible rule sets) is a powerset of the rule space. To obtain a maximum *a posteriori* (MAP) model within this solution space, Gibbs Sampling takes tens of thousands of iterations or more to converge even searching within a reduced space of only a couple of thousands of pre-mined and pre-selected rules [16, 29].

Here we propose an efficient inference algorithm that adopts the basic search procedure in simulated annealing. Given an objective function $p(R|S)$ over discrete search space of different rule sets and a temperature schedule function over time steps, $T^{[t]}$, a simulated annealing [13] procedure is a discrete time, discrete state Markov Chain where at step $t$, given the current state $R^{[t]}$, the next state $R^{[t+1]}$ is chosen by first proposing a neighbor and accepting it with probability that gradually decreases with time. In this framework, we also incorporate the following strategies for faster computation. 1) we use theoretical bounds for bounding the sampling chain to reduce computation. 2) instead of randomly proposing a neighboring solution, we aim to improve from the current solution by evaluating neighbors and pick the right one to move on to.

## 4.1   Theoretical bounds on MAP models

We exploit the model formulation to guide us in the search. We start by looking at MRS models with one rule removed. Removing a rule will yield a simpler model but may lower the likelihood. However, we can prove that the loss in likelihood is bounded as a function of the support. For a rule set $R$ and index $z$, we use $R_{\backslash z}$ to represent a set that contains all rules from $R$ except the $z^{\text{th}}$ rule, i.e., $R_{\backslash z} = \{r_m | r_m \in R, m \neq z\}$. Define

$$\Upsilon = \frac{\beta_-(N_+ + \alpha_+ + \beta_+ - 1)}{(N_- + \alpha_- + \beta_-)(N_+ + \alpha_+ - 1)},$$

where $N_+, N_-$ are the number of positive and negative examples, respectively. Notate the support of a rule as $\text{supp}(r) = \sum_n r(\mathbf{x}_n)$. Then the following holds:

**Lemma 1** *If* $\alpha_+ > \beta_+, \alpha_- > \beta_-$, *then* $p(\mathbf{y}|\mathbf{x}, R) \geq \Upsilon^{\text{supp}(z)} p(\mathbf{y}|\mathbf{x}, R_{\backslash z})$.

$\Upsilon$ is meaningful if $\Upsilon \leq 1$, otherwise this lemma means adding a rule always increases the conditional likelihood. This condition almost always holds since $\alpha_+ > \beta_+, \alpha_- > \beta_-$ and we do not set $\beta_+$ to a significantly large value. In practice it is recommended to set $\beta_+, \beta_-$ to 1.

We then introduce some notations that will be used later. Let $\mathcal{L}^*$ denote the maximum likelihood of data $S$, which is achieved when all data are classified correctly (this holds when $\alpha_+ > \beta_+$ and $\alpha_- > \beta_-$), i.e. $\text{TP} = N_+, \text{FP} = 0, \text{TN} = N_-$, and $\text{FN} = 0$, giving: $\mathcal{L}^* := B(N_+ + \alpha_+, \beta_+)B(N_- + \alpha_-, \beta_-)$. Let $v^{[t]}$ denote the best solution found until iteration $t$, i.e.,

$$v^{[t]} = \max_{\tau \leq t} p(R^{[\tau]}|S).$$

According to the prior model, containing too many rules penalizes the model due to the large complexity. Therefore, to hold a spot in the model, each rule needs to make enough "contribution" to the objective, i.e., capturing enough of the positive class, to cancel off the decrease in the prior. Therefore, we claim that the support of rule in the MAP model is lower bounded, and the bound becomes tighter as $v^{[t]}$ increases along the iterations.

**Theorem 2** *Take a data set $S$ and a MRS model with parameters*

$$H = \left\{ \alpha_M, \beta_M, \alpha_L, \beta_L, \alpha_+, \beta_+, \alpha_-, \beta_-, \{\theta_j\}_{j=1}^J \right\},$$

*where $H \in (\mathbb{N}^+)^{J+8}$. Define $R^* \in \arg\max_R p(R|S; H)$ and $M^* = |R^*|$. If $\alpha_M < \beta_M, \alpha_L < \beta_L, \alpha_+ > \beta_+, \alpha_- > \beta_-$ and $\Upsilon \leq 1$, we have:*

$$\forall r \in R^*, supp(r) \geq \left\lceil \frac{\log \frac{M^{[t]}\alpha_M \Omega}{M^{[t]} + \alpha_M - 1}}{\log \frac{1}{\Upsilon}} \right\rceil, \text{ and } M^{[t]} = \left\lfloor \frac{\log \mathcal{L}^* + \log p(\emptyset) - v^{[t]}}{\log \Omega} \right\rfloor,$$

*where* $\Omega = \frac{(\beta_M+1)(\beta_L+1)^{\alpha_L+1}\sum_{j=1}^J \theta_j}{\alpha_M \beta_L^{\alpha_L} \alpha_L \max(\theta)}$.

$p(\emptyset)$ is the prior of an empty set. $\mathcal{L}^*$ and $p(\emptyset)$ upper bound the conditional likelihood and prior, respectively. The difference between $\log \mathcal{L}^* + \log p(\emptyset)$ and $v^{[t]}$, the numerator in $M^{[t]}$, represents the room for improvement from the current solution $v^{[t]}$. The smaller the difference, the smaller the $M^{[t]}$. When we choose $\alpha_M = 1$, then the bound on support is reduced to

$$\text{supp}(r) \geq \left\lceil \frac{\log \Omega}{\log \frac{1}{\Upsilon}} \right\rceil.$$

We can control the bounds by changing parameters in $H$ to increase or decrease $\Omega$. As $\Omega$ increases, the bound $M^{[t]}$ decreases, which indicates a stronger preference for a simpler model with a smaller number of rules. Simultaneously, the lower bound for support increases, which is equivalent to reducing the search space. To increase $\Omega$, one can increase $\frac{\beta_M}{\alpha_M}$, which is the expected number of rules from the prior distribution, or increase $\frac{\beta_L}{\alpha_L}$, which is the expected number of items in each rule.

We incorporate the bound on the support in the search algorithm to check if a rule qualifies to be included.

## 4.2 Proposing step

Here we detail the proposing step at each iteration in the search algorithm. We simultaneously define the set of neighbors and the process to choose a neighbor to propose. A "next state" is proposed by first selecting an action to alter the current MRS and then choosing from "neighboring" models generated by that action. To improve the search efficiency, we do not perform a random action, but instead, we sample from misclassified examples to choose an action that can improve the current state $R^{[t]}$. If the misclassified example is positive, it means $R^{[t]}$ fails to "cover" it and therefore needs to increase the coverage by randomly choosing one of the following actions.

- *Add a value*: Choose a rule $r_m \in R^{[t]}$, a condition $c_k \in r_m$ and then a candidate value $v \in \mathcal{V}_{z_{mk}} \setminus \nu(c_k)$, then $c_k \leftarrow (z_{mk}, \nu(c_k) \cup v)$. $\nu(c_k)$ indicates the value(s) in condition $c_k$.
- *Remove a condition*: Choose a rule $r_m \in R^{[t]}$ and a condition $c_k \in r_m$, then $r_m = \{c_{k'} \in r_m | c_{k'} \neq c_k\}$
- *Add a rule*: Generate a new rule $r'$ where $\text{supp}(r')$ satisfies the bound in Theorem 2, $R^{[t+1]} \leftarrow R^{[t]} \cup r'$

where we use $\nu(\cdot)$ to access the feature in a condition.

On the other hand, if the misclassified example is negative, it means $R^{[t]}$ covers more than it should and therefore needs to reduce the coverage by randomly choosing one of the following actions.

- *Add a condition*: Choose a rule $r_m \in R^{[t]}$ first, choose a feature $j' \in \{1, \cdots, J\} \setminus z_m$ and then a set of values $V' \in \mathcal{V}_{j'}$, then update $r_m \leftarrow r_m \cup (j', V')$
- *Remove a rule*: Choose a rule $r_m \in R^{[t]}$, then $R^{[t+1]} = \{r \in R^{[t]} | r \neq r_m\}$

The above actions involve choosing a value, a condition, or a rule to perform the action on. Different choices result in different neighboring candidate models. To select one from them, we evaluate $p(\cdot|S)$ on every model. Then a choice is made between exploration (choosing a random model) and exploitation (choosing the best model). This randomness helps to avoid local minima and helps the Markov Chain to converge to a global optimum.

See the supplementary material for the complete algorithm.

## 5 Experimental Evaluation

We perform a detailed experimental evaluation of MRS models on simulated and real-world data sets. The first part of our experiments is designed to study the effect of hyperparameters on interpretability and predictive accuracy. The second part of the experiments compares MRS with classic and state-of-the-art benchmark baselines.

## 5.1 Accuracy & Interpretability Trade-off

We generate ten data sets of 100k observations with 50 arbitrary numerical features uniformly drawn from 0 to 1. For each data set, we construct a set of 10 rules by first drawing the number of conditions uniformly from 1 to 10 for each rule and then filling conditions with randomly selected features. Since the data are numeric, we generate a range for each feature by randomly selecting two values from 0 to 1, one as the lower boundary and the other as the upper boundary. These ten rules are the ground truth rule set denoted as $R^*$. Then we generate labels $\mathbf{y}$ from $R^*$: observations that satisfy $R^*$ are positive. Then each data set is partitioned into 75% training and 25% testing. To apply the MRS model, we discretize each feature into ten intervals and obtain a binary data set of size 100k by 500 on which we run the proposed model. We set entries in $\theta$ to 1, $\alpha_+ = \alpha_- = 100$ and $\beta_+ = \beta_- = 1$. Out of the four shape parameters $\alpha_M, \beta_M, \alpha_L, \beta_L$, we fix $\alpha_M, \alpha_L$ to 1 and only vary $\beta_M, \beta_L$. Larger values of $\beta_M, \beta_L$ indicate a stronger prior preference for simpler models. Let $\beta_M, \beta_L$ take values from $\{1, 10, 100, 1000, 10000\}$, giving a total of 25 sets of parameters. On each training data set, we run the MRS model with the 25 sets of parameters and then evaluate the output model on the test set. We repeat the process for ten data sets. Figure 1 shows the hold-out error, the number of conditions and the number of features used in the model. Each block corresponds to a parameter set. The values are averaged over ten data sets.

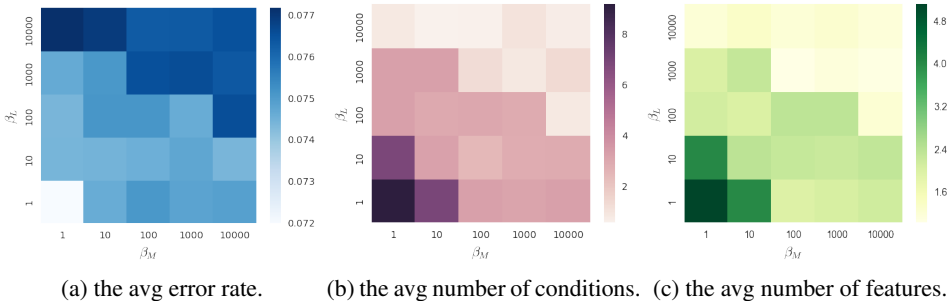

(a) the avg error rate.  (b) the avg number of conditions.  (c) the avg number of features.

Figure 1: Effect of shape parameters on predictive accuracy and interpretability.

The left-bottom corner represents models with the least constraint on complexity ($\beta_M = 1, \beta_L = 1$) and they achieve the lowest error but at the cost of the highest complexity and the largest feature set. As $\beta_M$ and $\beta_L$ increase, the model becomes less complex, with fewer conditions and fewer features, but at the cost of predictive accuracy. The right-top corner represents models with the strongest preference for simplicity: the smallest model with the largest error. The three figures show a clear pattern of the trade-off between interpretability and predictive accuracy.

## 5.2 Real World data sets

We then evaluate the performance of MRS on six real-world data sets from law enforcement, healthcare, and demography where interpretability is most desired. The data sets are publicly available at UCI Machine Learning Repository or ICPSR. Among these, medical data sets are especially suitable for MRS since many features such as diagnose categories have very high cardinalities.

Table 1: A summary of data sets

| data set | $N$ | $d$ | $Y = 1$ | Features |
|---|---|---|---|---|
| Juvenile Delinquency [23] | 4,023 | 69 | delinquency | exposure to violence, demo, etc |
| Credit card [34] | 30,000 | 24 | credit card default | gender, history of past payment, etc |
| Census [14] | 48,842 | 14 | income$\geq 50k$ | gender, age, occupation, etc |
| Recidivism | 11,645 | 106 | recidivism | conviction, employment, demo, etc |
| Hospital Readmission [27] | 100,000 | 55 | readmitted | diagnose history, symptoms, etc |
| In-hospital Mortality | 200,000 | 14 | death in hospital | diagnoses, medical history, etc |

**Baselines** We benchmark the performance of MRS against the following rule-based models for classification: Scalable Bayesian Rule Lists (SBRL) [33], Classification Based on Associations (CBA) [18], Repeated Incremental Pruning to Produce Error Reduction (Ripper) [5] and Bayesian Rule Sets (BRS) [31]. CBA and Ripper were designed to bridge the gap between association rule mining

and classification and thus focused mostly on optimizing for predictive accuracy. They are among the earliest and most-cited work on rule-based classifiers. On the other hand, BRS and SBRL, two recently proposed frameworks aim to achieve simpler models as well as high predictive accuracy. All of the four rule-based models use classical single value rules. Additionally, we would like to quantify the possible loss (if any) in predictive accuracy for gaining interpretability. Therefore, we also use two black-boxes, random forest and XGBoost to benchmark the performance without accounting for interpretability.

**Experimental Setup** We performed 5-fold cross validation for each method. In each fold, we set aside 20% of data during training for parameter tuning and used a grid search to locate the best set of parameters. We use R and python packages for the random forest, SBRL, CBA and Ripper [11] and use the publicly available code for BRS [1]. The MRS model has a set of hyperparameters $H_s, H_c$. We set entries in $\theta$ to 1, $\alpha_+ = \alpha_- = 100$ and $\beta_+ = \beta_- = 1$. $\alpha_M, \beta_M$ control the number of rules and $\alpha_L, \beta_L$ control lengths of rules. We set $\alpha_M, \alpha_L$ to 1 and vary $\beta_M, \beta_L$. We report in Table 2 the average test error, the average number of conditions in the output model, and the average number of unique features used in each model, computed from the 5 folds. The standard deviations are also reported.

Table 2: Evaluation of predictive performance and model complexity over 5-fold cross validation

| Task | Juvenile | | | Credit card | | | Census | | | Recidivism | | | Readmission | | | Mortality | | |
|---|---|---|---|---|---|---|---|---|---|---|---|---|---|---|---|---|---|---|
| Method | accuracy | $n_{cond}$ | $n_{feat}$ | accuracy | $n_{cond}$ | $n_{feat}$ | accuracy | $n_{cond}$ | $n_{feat}$ | accuracy | $n_{cond}$ | $n_{feat}$ | accuracy | $n_{cond}$ | $n_{feat}$ | F1 | $n_{cond}$ | $n_{feat}$ |
| Ripper | .88(.01) | 35(13) | 23(5) | .82(.01) | 23(8) | 12(2) | .84(.01) | 67(11) | 7(0) | .78(.00) | 78(18) | 32(4) | .58(.01) | 35(9) | 12(1) | .26(.01) | 115(6) | 9(1) |
| CBA | .88(.01) | 27(22) | 18(12) | .80(.01) | 35(3) | 6(0) | .79(.01) | 13(12) | 6(2) | .72(.01) | 87(25) | 27(5) | .61(.01) | 39(10) | 13(1) | .28(.02) | 435(18) | 10(2) |
| SBRL | .88(.01) | 10(2) | 9(2) | .82(.00) | 15(2) | 10(2) | .82(.00) | 32(2) | 10(1) | .75(.00) | 10(1) | 9(1) | .61(.01) | 21(1) | 7(1) | .30(.01) | 6(1) | 4(1) |
| BRS | .88(.01) | 21(4) | 11(3) | .81(.01) | 17(2) | 8(2) | .79(.01) | 33(11) | 11(2) | .73(.01) | 16(11) | 8(3) | .59(.01) | 9(11) | 5(3). | .39(.01) | 10(1) | 4(0) |
| **MRS** | .89(.00) | 18(3) | 6(2) | .82(.01) | 10(7) | 5(3) | .80(.00) | 14(8) | 5(3) | .74(.02) | 6(3) | 3(1) | .60(.00) | 6(3) | 3(0) | .39(.00) | 6(2) | 3(1) |
| | $n_{val}$: | 19(1) | | $n_{val}$: | 13(5) | | $n_{val}$: | 29(17) | | $n_{val}$: | 8(3) | | $n_{val}$: | 8(4) | | $n_{val}$: | 8(2) | |
| RF | .90(.00) | – | – | .82(.00) | – | – | .86(.00) | – | – | .74(.00) | | | .61(.00) | – | – | .41(.01) | – | – |
| XGBoost | .91(.01) | – | – | .83(.01) | – | – | .87(.01) | – | – | .75(.05) | – | – | .60(.00) | – | – | .41(.02) | – | – |

**Results** We evaluate the predictive performance and interpretability performance by measuring three metrics: i) the accuracy on the test set (we report F1 score for the mortality data set since it is highly unbalanced), ii) the total number of conditions in the output model (for MRS models, we also report the total number of values), and iii) the average number of unique features in the model. MRS achieves consistently competitive predictive accuracy using significantly fewer conditions and fewer features. On data sets credit card and mortality, MRS is the best performing model: highest accuracy, smallest complexity, and fewest features. On juvenile data set, MRS achieves the highest accuracy while using the second smallest number of conditions. On readmission data set, MRS loses slightly in accuracy compared to CBA and SBRL but only uses 6 conditions while CBA used 39 and SBRL used 21. In summary, MRS models use the fewest conditions on five out of six data sets. They use the smallest number of features on all six data sets, even for juvenile data set where MRS has more conditions than SBRL model but still wins in the number of features.

We show an MRS model learned from data set juvenile to inspect if the grouping of categories is meaningful. It consists of two rules, and if a teenager satisfies either of them, then the model predicts the teenager will conduct delinquency in the future. In this data set, features are questions and feature values are choices for the questions.

1: [Have your friends ever hit or threatened to hit someone without any reason? = "All of them" or "Not sure" or "Refused to Answer"]

2: [Have your friends purposely damaged or destroyed property that did not belong to them? = "All of them" or "Most of them" or "Some of them"] *AND* [Did any of your family members use hard drugs? = "Yes"] *AND* ["Has any of your family members or friends ever beat you up with their fists so hard that you were hurt pretty bad? = "Yes"]

It is interesting to notice that MRS grouped three values in the first rule together and the three values in the first condition in the second rule. Grouped values are considered interchangeable by the model. It is intuitive to explain the grouping with common sense. People avoid answering when they feel alerted or uncomfortable with the question [4, 32]. In this case, this question concerns the privacy of their friends, making people more reserved and hesitant to provide a definite answer. So they would rather say they are not sure or refuse to answer than directly say yes.

### 5.3 Interpretability Evaluation by Humans

To further evaluate the model interpretability, in addition to quantitively measuring the size of the model, we would like to understand how quickly and how correctly humans understand a machine learning model. We designed a short survey and sent it to a group of 70 undergraduate students. The survey was designed as an online quiz with credit to motivate students to do it as accurately as possible. The students have been enrolled in a machine learning class for a couple of weeks and have some knowledge about predictive models.

We chose to show models built from data set "credit card" since output models are smallest compared to other data sets, so it's easier for humans to understand. The students were asked to use the models to make predictions on given instances. Every method has five models, each from one of the five folds. Therefore, each student was shown with one model for every one of the five methods. The survey first taught them how to use a model with instructions and an example, and then asked them to use the model to make predictions on two instances. Their answers and response time were recorded.

Since all competing methods are rule-based models, it is important that students understand the notion of rules before working with any of the models. Therefore, we designed a screening question on rules and students can only proceed with the survey if they answered the question correctly. 66 students passed the test.

We report in Figure 2 the accuracy and response time of each method averaged over five folds. Note that response time refers to the total time for understanding the model and using the model to predict two instances. Accuracy was evaluated against the predictions of a model, not the true labels. Methods MRS and BRS achieve the highest accuracy, and SBRL achieves the lowest accuracy. We hypothesize this is because SBRL uses an ordered set of rule connected by "else-if" which makes it a little more difficult to understand compared to un-ordered rules in the other methods. For the response time, MRS uses a significantly small amount of time, less than half of that of CBA and Ripper, due to the Bayesian prior to favor small models and a concise presentation allowing multiple conditions in a rule. BRS also takes a very short time, a bit longer than MRS, followed by SBRL. MRS, BRS, and SBRL all have a Bayesian component to favor small models while CBA and Ripper do not, thus taking significantly longer to understand and use.

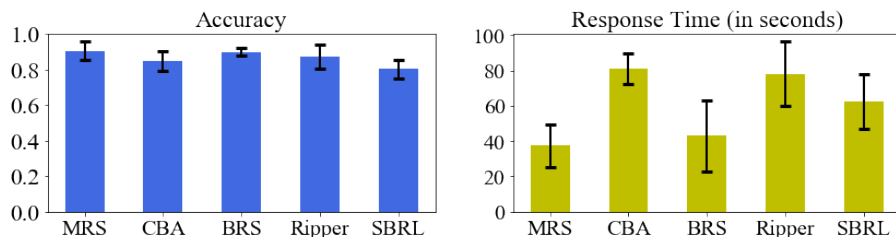

Figure 2: Effect of shape parameters on predictive accuracy and interpretability

## 6 Conclusions

We proposed a Multi-value Rule Set (MRS) which provides a more concise and feature-efficient model form to classify and explain. We developed an inference algorithm that incorporates theoretically grounded bounds to reduce computation. Compared with state-of-the-art rule-based models, MRS showed competitive predictive accuracy while achieving a significant reduction in complexity and feature sets, thus improving the interpretability, demonstrated by human evaluation. A major contribution is that we demonstrated the possibility of using fewer features without hurting too much (if any) predictive performance.

Note that we do not claim that multi-value rules are more interpretable than single-value rules since it is well-known that interpretability comes in different forms for different domains. However, our model provides a more flexible solution for interpretable models since, after all, a single-value rule is just a special case of multi-value rules. We believe the potential in the proposed multi-value rules is not only limited to MRS. They can be adopted in other rule-based models.

**Code**: The MRS code is available at `https://github.com/wangtongada/MRS`.

## Footnotes

[1] https://github.com/wangtongada/BOA

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
