[Supplementary Material · supplementaryv1.pdf]

# Supplementary Material for Multi-value Rule Sets for Interpretable Classification with Feature-Efficient Representations

**Tong Wang**
Tippie School of Business
University of Iowa
Iowa City, IA 52242
`tong-wang@uiowa.edu`

## 1 Proofs

**Proof 1** *(of Theorem 1) Since $R$ and $R'$ differs only in two features in one rule, and has the same size of the rules, so we only need to compare the probability of feature selection for the rule. Let $l'_{mj_1}, l'_{mj_2}$ represent the number of items taking feature $j_1$ and $j_2$ in the $m$-th rule in $R'$. $l'_{mj_1} = l_{mj_1} + 1, l'_{mj_2} = l_{mj_2} - 1$*

$$\frac{p(r_m|L_m, \theta)}{p(r'_m|L_m, \theta)} = \frac{\Gamma(l_{mj_1} + \theta_{j_1})}{\Gamma(l'_{mj_1} + \theta_{j_1})} \frac{\Gamma(l_{mj_2} + \theta_{j_2})}{\Gamma(l'_{mj_2} + \theta_{j_2})}$$
$$= \frac{l_{mj_2} + \theta_{j_2} - 1}{l_{mj_1} + \theta_{j_1}}$$
$$< 1.$$

**Proof 2** *(of Lemma 1) Given a MRS model $R$, let TP, FP, TN and FN be the number of true positives, false positives, true negatives and false negatives in $S$ classified by $R$. The conditional likelihood is*

$$p(\mathbf{y}|\mathbf{x}, R = \frac{B(TP + \alpha_+, FP + \beta_+)}{B(\alpha_+, \beta_+)} \frac{B(TN + \alpha_-, FN + \beta_-)}{B(\alpha_-, \beta_-)}.$$

*We then compute the likelihood for model $R_{\setminus z}$. The most extreme case is when rule $z$ is an $100\%$ accurate rule that applies only to real positive data points and those data points satisfy only $z$. Therefore once removing it, the number of true positives decreases by $supp(z)$ and the number of false negatives increases by $supp(z)$. That is,*

$$p(\mathbf{y}|\mathbf{x}, R_{\setminus z}) \geq \frac{B(TP - supp(z) + \alpha_+, FP + \beta_+)}{B(\alpha_+, \beta_+)} \frac{B(TN + \alpha_-, FN + supp(z) + \beta_-)}{B(\alpha_-, \beta_-)}$$
$$= p(\mathbf{y}|\mathbf{x}, A) \cdot g(supp(z)), \tag{1}$$

*where*

$$g(supp(z)) = \frac{\Gamma(TP + \alpha_+ - supp(z))}{\Gamma(TP + \alpha_+)} \frac{\Gamma(TP + FP + \alpha_+ + \beta_+)}{\Gamma(TP + FP + \alpha_+ + \beta_+ - supp(z))}$$
$$\frac{\Gamma(FN + \beta_- + supp(z))}{\Gamma(FN + \beta_-)} \frac{\Gamma(TN + FN + \alpha_- + \beta_-)}{\Gamma(TN + FN + \alpha_- + \beta_- + supp(z))}. \tag{2}$$

*Now we break down $g(supp(z))$ to find a lower bound for it. The first two terms in (2) become*

$$\frac{\Gamma(TP + \alpha_+ - supp(z))}{\Gamma(TP + \alpha_+)} \frac{\Gamma(TP + FP + \alpha_+ + \beta_+)}{\Gamma(TP + FP + \alpha_+ + \beta_+ - supp(z))}$$

$$= \frac{(TP + FP + \alpha_+ + \beta_+ - supp(z)) \ldots (TP + FP + \alpha_+ + \beta_+ - 1)}{(TP + \alpha_+ - supp(z)) \ldots (TP + \alpha_+ - 1)}$$

$$\geq \left(\frac{TP + FP + \alpha_+ + \beta_+ - 1}{TP + \alpha_+ - 1}\right)^{supp(z)}$$

$$\geq \left(\frac{N_+ + \alpha_+ + \beta_+ - 1}{N_+ + \alpha_+ - 1}\right)^{supp(z)}. \tag{3}$$

*Equality holds in (3) when $TP = N_+, FP = 0$. Similarly, the last two terms in (2) become*

$$\frac{\Gamma(FN + \beta_- + supp(z))}{\Gamma(FN + \beta_-)} \frac{\Gamma(TN + FN + \alpha_- + \beta_-)}{\Gamma(TN + FN + \alpha_- + \beta_- + supp(z))}$$

$$= \frac{(FN + \beta_-) \ldots (FN + \beta_- + supp(z) - 1)}{(TN + FN + \alpha_- + \beta_-) \ldots (TN + FN + \alpha_- + \beta_- + supp(z) - 1)}$$

$$\geq \left(\frac{FN + \beta_-}{FN + TN + \alpha_- + \beta_-}\right)^{supp(z)}$$

$$\geq \left(\frac{\beta_-}{N_- + \alpha_- + \beta_-}\right)^{supp(z)}. \tag{4}$$

*Equality in (4) holds when $TN = N_-, FN = 0$. Combining (1), (2), (3) and (4), we obtain*

$$p(\mathbf{y}|\mathbf{x}, R_{\setminus z}) \geq \left(\frac{N_+ + \alpha_+ + \beta_+ - 1}{N_+ + \alpha_+ - 1} \frac{\beta_-}{N_- + \alpha_- + \beta_-}\right)^{supp(z)} \cdot p(\mathbf{y}|\mathbf{x}, R)$$

$$= \Upsilon^{supp(z)} p(\mathbf{y}|\mathbf{x}, R). \tag{5}$$

**Proof 3** *(of Theorem 2)*
**Step 1** *We first prove the upper bound on $M^*$. Since $R^* \in \arg\max_R p(R|S)$, $p(R^*|S) \geq v^{[t]}$, i.e.,*

$$\log p(S|R^*) + \log p(R^*) \geq v^{[t]}. \tag{6}$$

*Since $p(S|R^*) \leq \mathcal{L}^*$, we only need to upper bound $p(R^*)$.*

*The prior probability of $R^*$ depends on the number of rules $M^*$, the number of items in each rule, which we denote as $L_m, m \in \{1, ..., M^*\}$, and the number of items containing each feature, denoted as $\{l_{mj}\}_j$, so the prior probability for $R^*$ is*

$$p(R^*) \propto p(M^*|\alpha_M, \beta_m) \prod_{m=1}^{M} p(L_m|\alpha_L, \beta_L) p(r_m|L_m, \theta) \tag{7}$$

*where*

$$p(M^*|\alpha_M, \beta_M) = \frac{\Gamma(M^* + \alpha_M)}{\Gamma(M^* + 1)\Gamma(\alpha_M)} \left(\frac{1}{\beta_M + 1}\right)^M \left(\frac{\beta_M}{\beta_M + 1}\right)^{\alpha_M}$$

$$\leq \frac{\alpha_M(\alpha_M + 1) \cdots (\alpha_M + M^* - 1)}{1 \cdot 2 \cdots M^*} \left(\frac{1}{\beta_M + 1}\right)^{M^*} \left(\frac{\beta_M}{\beta_M + 1}\right)^{\alpha_M}$$

$$= \left(\frac{\alpha_M}{\beta_M + 1}\right)^{M^*} \left(\frac{\beta_M}{\beta_M + 1}\right)^{\alpha_M} \tag{8}$$

*Similarly, for each $m \in \{1, ..., M\}$,*

$$p(L_m|\alpha_L, \beta_L) \leq \left(\frac{\alpha_L}{\beta_L + 1}\right)^{L_m} \left(\frac{\beta_L}{\beta_L + 1}\right)^{\alpha_L}$$

*We also upper bound $p(r_m|L_m, \theta)$:*

$$p(z_m|L_m, \theta) = \frac{\Gamma(\sum_{j=1}^J \theta_j)}{\Gamma(L_m + \sum_{j=1}^J \theta_j)} \prod_{j=1}^J \frac{\Gamma(l_{mj} + \theta_j)}{\Gamma(\theta_j)}$$

$$\leq \frac{\Gamma(\sum_{j=1}^J \theta_j)\Gamma(L_m + \max(\theta))}{\Gamma(L_m + \sum_{j=1}^J \theta_j)\Gamma(\max(\theta))}$$

$$= \frac{\max(\theta)(\max(\theta)+1)\cdots(\max(\theta)+L_m-1)}{(\sum_{j=1}^J \theta_j)(\sum_{j=1}^J \theta_j + 1)\cdots(\sum_{j=1}^J \theta_j + L_m - 1)}$$

$$\leq \left( \frac{\max(\theta)+L_m-1}{\sum_{j=1}^J \theta_j + L_m - 1} \right)^{L_m}$$

$$\leq \frac{\max(\theta)}{\sum_{j=1}^J \theta_j} \tag{9}$$

*(9) is because $\left( \frac{\max(\theta)+L_m-1}{\sum_{j=1}^J \theta_j + L_m - 1} \right)^{L_m}$ decreases monotonically with $L_m$. Therefore*

$$p(R^*) \leq \left( \frac{\alpha_M}{\beta_M + 1} \right)^{M^*} \left( \frac{\beta_M}{\beta_M + 1} \right)^{\alpha_M} \prod_{m=1}^M \left( \frac{\alpha_L}{\beta_L + 1} \right)^{L_m} \frac{\max(\theta)}{\sum_{j=1}^J \theta_j} \left( \frac{\beta_L}{\beta_L + 1} \right)^{\alpha_L}$$

$$\leq \left( \frac{\alpha_M}{\beta_M + 1} \right)^{M^*} \left( \frac{\beta_M}{\beta_M + 1} \right)^{\alpha_M} \prod_{m=1}^M \frac{\alpha_L}{\beta_L + 1} \frac{\max(\theta)}{\sum_{j=1}^J \theta_j} \left( \frac{\beta_L}{\beta_L + 1} \right)^{\alpha_L} \tag{10}$$

*(10) follows because $\alpha_L < \beta_L$. Since $p(\emptyset) = \left( \frac{\beta_M}{\beta_M+1} \right)^{\alpha_M}, \Omega = \frac{(\beta_M+1)(\beta_L+1)^{\alpha_L+1}\sum_{j=1}^J \theta_j}{\alpha_M \beta_L^{\alpha_L} \alpha_L \max(\theta)}$, and $\Omega > 1$, we have*

$$p(R^*) = p(\emptyset) \left( \frac{\alpha_M \beta_L^{\alpha_L} \alpha_L \max(\theta)}{(\beta_M + 1)(\beta_L + 1)^{\alpha_L+1}\sum_{j=1}^J \theta_j} \right)^{M^*}$$

$$= p(\emptyset)(\frac{1}{\Omega})^{M^*} \tag{11}$$

*Now we apply (6) combining with (11), we get*

$$\log \mathcal{L}^* + \log p(\emptyset) + M^* \log \frac{1}{\Omega} \geq v^{[t]} \tag{12}$$

*Then solving for $M^*$ yields:*

$$M^* \leq M^{[t]} = \left\lfloor \frac{\log \mathcal{L}^* + \log p(\emptyset) - v^{[t]}}{\log \Omega} \right\rfloor. \tag{13}$$

*where we use $M^{[t]}$ to denote the upper bound derived at time $t$.*

***Step 2:*** *Now we prove the lower bound on the support. We would like to prove that a MAP model does not contain rules of support less than a threshold. To show this, we prove that if any rule $z$ has support smaller than some constant, then removing it yields a better objective, i.e.,*

$$p(R^*|S) \leq p(R^*_{\backslash z}|S). \tag{14}$$

*Our goal is to find the constant such that this inequality holds. We relate $P(R_{\backslash z})$ with $P(R)$. We multiply $P(R_{\backslash z}; \theta)$ with 1 in disguise to relate it to $P(R)$:*

$$p(R^*_{\backslash z}) = p(M^* - 1; \alpha_M, \beta_M) \prod_{m \neq z}^{M^*} p(L_m; \alpha_L, \beta_L)p(r_m|L_m, \theta)$$

$$= \frac{p(M^* - 1; \alpha_M, \beta_M)}{p(M^*; \alpha_M, \beta_M)p(L_z; \alpha_L, \beta_L)p(r_m|L_m; \alpha)}p(A)$$

$$\geq \frac{M^*(\beta_M + 1)}{(M^* + \alpha_M - 1)} \left( \frac{\beta_L + 1}{\beta_L} \right)^{\alpha_L} \left( \frac{\beta_L + 1}{\alpha_L} \right)^{L_z} \left( \frac{\sum_{j=1}^J \theta_j}{\max(\theta)} \right) p(R)$$

$$\geq \frac{M^*(\beta_M + 1)(\beta_L + 1)^{\alpha_L + 1}(\sum_{j=1}^{J} \theta_j)}{(M^* + \alpha_M - 1)\beta_L^{\alpha_L}\alpha_L \max(\theta)} p(A)$$

$$= \frac{M^*\alpha_M}{M^* + \alpha_M - 1}\Omega p(R). \tag{15}$$

*From Theorem 1 we also have*

$$p(\mathbf{y}|\mathbf{x}, R_{\setminus z}) \geq \Upsilon^{supp(z)}p(\mathbf{y}|\mathbf{x}, R). \tag{16}$$

*Then combining (15) with (16), the joint probability of S and $R_{\setminus z}$ is lower bounded by*

$$p(R_{\setminus z}) \geq \frac{M^*\alpha_M}{M^* + \alpha_M - 1} \cdot \Omega\Upsilon^{supp(z)}p(R|S).$$

*In order to get $p(R_{\setminus z}|S) \geq p(R|S)$, we need*

$$\frac{M^*\alpha_M}{M^* + \alpha_M - 1} \cdot \Omega\Upsilon^{supp(z)} \geq 1,$$

*i.e.,*

$$\Upsilon^{supp(z)} \geq \frac{M^* + \alpha_M - 1}{M^*\alpha_M\Omega} \geq \frac{M^{[t]} + \alpha_M - 1}{M^{[t]}\alpha_M\Omega},$$

*We have $\Upsilon \leq 1$, thus*

$$supp(z) \leq \frac{\log \frac{M^{[t]}\alpha_M\Omega}{M^{[t]}+\alpha_M-1}}{\log \frac{1}{\Upsilon}}. \tag{17}$$

*Therefore, for any rule $z$ in a MAP model $R^*$,*

$$supp(z) \geq \left\lceil \frac{\log \frac{M^{[t]}\alpha_M\Omega}{M^{[t]}+\alpha_M-1}}{\log \frac{1}{\Upsilon}} \right\rceil. \tag{18}$$

## 2 Inference Algorithm

Below we present the full inference algorithm

---
**Algorithm 1** Inference algorithm.

---
**procedure** SIMULATED ANNEALING($N_{\text{iter}}$)
    $R^{[0]} \leftarrow$ a randomly generated rule set
    **for** $t = 0 \rightarrow N_{\text{iter}}$ **do**
        $(\mathbf{x}_k, y_k) \leftarrow$ a random example drawn from data points misclassified by $R^{[t]}$
        **if** $y_k = 1$ **then**
$$R^{[t+1]} \leftarrow \begin{cases} \text{AddValue}(R^{[t]}), \text{ with probability } \frac{1}{3} \\ \text{RemoveCondition}(R^{[t]}), \text{ with probability } \frac{1}{3} \\ \text{AddRule}(R^{[t]}), \text{ with probability } \frac{1}{3} \text{ (using Theorem 2)} \end{cases}$$
        **else**
$$R^{[t+1]} \leftarrow \begin{cases} \text{AddCondition}(R^{[t]}), \text{ with probability } \frac{1}{2} \\ \text{RemoveRule}(R^{[t]}), \text{ with probability } \frac{1}{2} \end{cases}$$
        $R_{\max} \leftarrow \arg\max(p(R_{\max}|S), p(R^{[t+1]}|S))$ (Check for improved optimal solution)
        $\alpha = \min\left\{1, \exp\left(\frac{p(R^{[t+1]}|S) - p(R^{[t]}|S)}{T^{[t]}}\right)\right\}$ (Probability of an annealing move)
        $R^{[t+1]} \leftarrow R^{[t+1]}$, with probability $\alpha$
    **end for**
    return $A_{\max}$
**end procedure**

---