[Reviews · NeurIPS 2018]

Reviewer 1



The paper proposes learning sets of decision rules that can express the disjunction of feature values in atoms of the rules, for example, IF color == yellow OR red, THEN stop. The emphasis is on interpretability, and the paper argues that these multi-value rules are more interpretable than similarly trained decision sets that do not support multi-value rules. Following prior work, the paper proposes placing a prior distribution over the parameters of the decision set, such as the number of rules and the maximum number of atoms in each rule. The paper derives bounds on the resulting distribution to accelerate a simulated annealing learning algorithm. Experiments show that multi-value rule sets are as accurate as other classifiers proposed as interpretable model classes, such as Bayesian rule sets on benchmark decision problems. The paper argues that multi-value rule sets are more interpretable on this problem. The paper is sound and clearly written. It seems a straightforward extension of work on Bayesian rule sets (Wang et al. ICDM 2016). My main questions are about its evaluation. The significant feature of the work is the use of multi-valued rules which are motivated by interpretability, which is tricky to evaluate. No user study or human evaluation was conducted on the learned models. The primary comparison is conducting by counting the number of rules and features used in the learned models. Learned multi-valued rules have fewer rules and features than other models, but no information on the number of feature values per rule is provided. I agree that a rule like IF A == V1 or V2, THEN... is more interpretable than two rules IF A == V1, THEN... and IF A == V2, THEN... (2 rules vs. 1 rule) but I do not agree that it is even as interpretable than just the rule IF A == V1 (1 rule with 2 feature values vs. 1 rule with 1 feature value). It seems that there is an unexplored tradeoff space here, and only partial information is provided on where each model is in in that space. UPDATE: I appreciate how thoroughly the authors responded to my review. I am raising my score to a 7.

Reviewer 2



Summary: This paper presents a generative rule-learning model that uses a two-stage learning process to first generate model structure and then assign features and values into the model structure. Two critical properties of the proposed approach are that it has an inductive bias to use few features and cluster discrete feature values to improve model support. Although performance on several datasets is close to the popular rule induction methods, the proposed method produces rules with fewer conditions and requiring fewer features. Pros: - Bayesian, probabilistic approach contrasts with many of the deterministic methods of rule induction - Focus on using few features and clustering values is practical for explainability, and efficient evaluation. - Intelligent sampling approaches to improve scalability of complex inference Cons: - Does not concretely discuss scalability of the method w.r.t. features, values, instances, etc. - Ignores related work in structure learning and lifted inference in probabilistic modeling that has similar aims - Evaluation is restricted to a few, smaller datasets - could be more extensive Quality: 3/5 - identifies a problem, introduces a novel solution, and tests it Clarity: 4/5 - Well-written and clear Originality: 3/5 - Not entirely familiar with this subcommunity, but the ideas seem novel in rule learning, but structure learning work already employs many of the same ideas. Significance: 3/5 - Without better bounds and experiments on scaling, don't know if this is really a tractable model Detailed comments: This is an interesting paper that uses a different approach from prior work on rule induction -- a fully generative model. The model itself seems a straightforward Bayesian approach to rule learning, but some elements such as feature clustering and efficient sampling from misclassifications are clever. I have several misgivings about this paper. The most serious is scalability: is this a method that could conceivably be used in real-world settings where there are more instances, more features, many discrete options (or continuous values), and few training examples? The complexity of the inference suggests that this method could face many hurdles in those real-world settings. The second issue I'd note is that this paper reminded me of the work in the probabilistic modeling community on lifted inference (see the tutorial "Lifted Probabilistic Inference in Relational Models" as a good starting point). The goals and techniques of this community, parsimonious models that maximize likelihood while bounding model complexity, seem very similar to this technique. Comparing against these approaches, in both theoretically and empirically would help contextualize this work more broadly. Finally, the experiments seem fairly constricted for the level of generality of this method. Classical rule learning papers demonstrated performance on ~40 datasets, so limiting results to 3 datasets calls into question some of the generality of the approach. Additional experiments would also bolster claims of scalability.

Reviewer 3



Review: The paper proposes a bayesian rule based model that allows for the inclusion of multiple values of the same feature inside a condition. The model is composed of a set of rules, and an example is classified as positive if it is covered by at least one rule. The generative model has knobs to encourage three interpretability-related properties : - Small number of rules - Small number of conditions within each rule - For each rule, small number of features used (i.e. conditions are clustered into few features) The improvements of MRS over other rule-based frameworks are compelling. Rule sets that use less features are easier to grasp, and rules with multiple values for the same feature are more intuitive and easier to grasp than the same condition fragmented over multiple conditions. The paper is mostly clear, but there is a lot of notation. Some suggestions to improve clarity: - Maybe draw the generative model - Change the notation of multiple values to [state \in {California, Texas, Arizona, Oregon}] rather than [state =California or Texas or Arizona or Oregon], or in the very least change the spacing around the = - Drop the second _m in {L_m}_m, {z_m}_m, etc. - The subscript plus and minus in 4.1 makes the equations very hard to read Comments / questions: - Is there a reason for not constraining alpha_+ > beta_+ and alpha_- > beta_- in the formulation? This would avoid the repetition of this condition around Lemma 1. - It seems that removing a condition reduces the coverage, while adding a condition increases the coverage. These are reversed in 4.2. - The comparison of number of conditions in Table 1 does not seem entirely fair, as the conditions for MRS are much longer, unless the authors are considering each value in a condition as a separate condition for this count. - The discussion in lines 326 - 330 does not seem to make sense given that "All of them" is a value included in the condition. This value breaks the intuition presented. - It's not clear how the authors selected the best model in cross validation after doing grid search. Was the selection done in a systematic or ad hoc way? Overall, I think this is an interesting interpretable model, with the caveats that it only works for binary classification (and the extensions to more labels is not obvious) and that there are many hyperparameters that need tuning. The results dot not present significant improvements over prior art in terms of accuracy, but there seem to be some improvements in terms of interpretability (although the comparison may not be fair, see comment above). Typos: 26: primitive concept level -> primitive concepts 51: "an examples" -> examples, "fo example" -> for example 55: for its appealing property (weird phrasing) 68: remove 'the' 73: remove stop before (Millions) 75: 'and it was done…' weird phrasing 78: various work 79: conjunction connector -> the conjunction connector 81: conjunction/literal -> conjunction / literal (add space) 94: 'simply adapt' -> be adapted 306: uniqeu -> unique 310: 'accuracy 67 conditions' - something is missing here 316: two stops after 'meaningful' 342: MARS -> MRS Supplementary line 6: MARS -> MRS # Comments on feedback: - Thanks for addressing the concern about fair comparison - I think that the user study makes the submission stronger, and so do the added results in the new Table 1. I've changed my score to reflect that. - I had understood that 'all of them' is a value like the others. I am question the intuition presented, which states that people prefer to say they are not sure or refuse to answer. This intuition does not fit well with the inclusion of 'all of them' on the right hand side of the rule.